# LIGHT-WEIGHT PROBING OF UNSUPERVISED REPRESENTATIONS FOR REINFORCEMENT LEARNING

## ABSTRACT

Unsupervised visual representation learning offers the opportunity to leverage large corpora of unlabeled trajectories to form useful visual representations, which can benefit the training of reinforcement learning (RL) algorithms. However, evaluating the fitness of such representations requires training RL algorithms which is computationally intensive and has high variance outcomes. To alleviate this issue, we design an evaluation protocol for unsupervised RL representations with lower variance and up to 600x lower computational cost. Inspired by the vision community, we propose two linear probing tasks: predicting the reward observed in a given state, and predicting the action of an expert in a given state. These two tasks are generally applicable to many RL domains, and we show through rigorous experimentation that they correlate strongly with the actual downstream control performance on the Atari100k Benchmark. This provides a better method for exploring the space of pretraining algorithms without the need of running RL evaluations for every setting. Leveraging this framework, we further improve existing self-supervised learning (SSL) recipes for RL, highlighting the importance of the forward model, the size of the visual backbone, and the precise formulation of the unsupervised objective. Code will be released upon acceptance.

## 1    INTRODUCTION

Learning visual representations is a critical step towards solving many kinds of tasks, from supervised tasks such as image classification or object detection, to reinforcement learning (RL). Ever since the early successes of deep reinforcement learning (Mnih et al., 2015), neural networks have been widely adopted to solve pixel-based reinforcement learning tasks such as arcade games (Bellemare et al., 2013), physical continuous control (Todorov et al., 2012; Tassa et al., 2018), and complex video games (Synnaeve et al., 2018; Oh et al., 2016). However, learning deep representations directly from rewards is a challenging task, since this learning signal is often noisy, sparse and delayed.

With ongoing progress in unsupervised visual representation learning for vision tasks (Zbontar et al., 2021; Chen et al., 2020a;b; Grill et al., 2020; Caron et al., 2020; 2021), recent efforts have likewise applied self-supervised techniques and ideas to improve representation learning for RL. Some promising approaches include supplementing the RL loss with self-supervised objectives (Laskin et al., 2020; Schwarzer et al., 2021a), or first pre-training the representations on a corpus of trajectories (Schwarzer et al., 2021b; Stooke et al., 2021). However, the diversity in the settings considered, as well as the self-supervised methods used, make it difficult to identify the core principles of successful self-supervised methods in RL. Moreover, estimating the performance of RL algorithms is notoriously challenging (Henderson et al., 2018; Agarwal et al., 2021): it often requires repeating the same experience with a different random seed, and the high CPU-to-GPU ratio is a compute requirement of most online RL methods that is inefficient for typical research compute clusters. This hinders systematic exploration of the many design choices that characterize SSL methods.

In this paper, we strive to provide a reliable and lightweight evaluation scheme for unsupervised visual representation in the context of RL. Inspired by the vision community, we propose to evaluate the representations using linear probing, by training a linear prediction head on top of frozen features. We devise two probing tasks that we deem widely applicable: predicting the reward in a given state, and predicting the action that would be taken by a fixed policy in a given state (for example that of an expert). We stress that these probing tasks are only used as a means of evaluation. Because

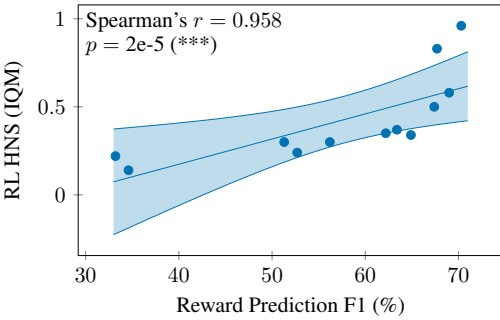 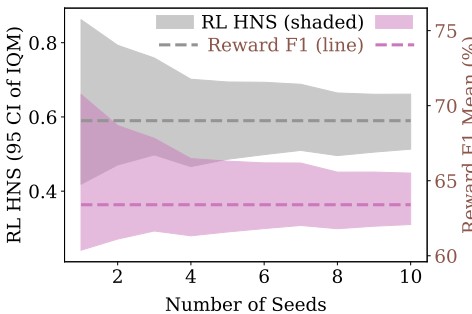

Figure 1: **Left**: Correlation between the SSL representations' abilities to linearly predict the presence of reward in a given state, versus RL performance using the same representations, measured as the interquartile mean of the human-normalized score (HNS) over 9 Atari games. Each point denotes a separate SSL pretraining method. A linear line of best fit is shown with 95 confidence interval. We compute Spearman's rank correlation coefficient (Spearman's r) and determine its statistical significance using permutation testing (with $n = 50000$). **Right:** When comparing two models, the reward probing score can give low variance reliable estimates of RL performance, while direct RL evaluation may require many seeds to reach meaningful differences in mean performance.

very little supervised data is required, they are particularly suitable for situations where obtaining the expert trajectories or reward labels is expensive. Through thorough experimentation, we show that the performance of the SSL algorithms (in terms of their downstream RL outcomes) correlates with the performance in both probing tasks with statistically significant (p<0.001) Spearman's rank correlation, making them particularly effective proxies. Given the vastly reduced computational burden of linear evaluations, we argue that it enables much easier and straightforward experimentation of SSL design choices, paving the way for a more systematic exploration of the design space.

Finally, we leverage this framework to systematically assess some key attributes of SSL methods. First off, we explore the utility and role of learning a forward model as part of the self-supervised objective. We investigate whether its expressiveness matters and show that equipping it with the ability to model uncertainty (through random latent variable) significantly improves the quality of the representations. Next, we identify several knobs in the self-supervised objective, allowing us to carefully tune the parameters in a principled way. Finally, we confirm the previous finding (Schwarzer et al., 2021b) that bigger architectures, when adequately pre-trained, tend to perform better.

Our contributions can be summarized as follows:

- Design of a rigorous and efficient SSL evaluation protocol in the context of RL
- Empirical demonstration that this evaluation scheme correlates with downstream RL performance
- Systematic exploration of design choices in existing SSL methods.

## 2 RELATED WORK

### 2.1 REPRESENTATION LEARNING

There has recently been a surge in interest and advances in the domain of self-supervised learning in computer vision. Some state-of-art techniques include contrastive learning methods SimCLR, MoCov2 (Chen et al., 2020a;b); clustering methods SwAV (Caron et al., 2020); distillation methods BYOL, SimSiam, OBoW (Grill et al., 2020; Chen and He, 2021; Gidaris et al., 2020); and information maximization methods Barlow Twins and VicReg (Zbontar et al., 2021; Bardes et al., 2021).

These advances have likewise stimulated development in representation learning for reinforcement learning. A line of work includes unsupervised losses as an auxiliary objective during RL training to improve data efficiency. Such objective can be contrastive (Laskin et al., 2020; Zhu et al., 2020)

or non-contrastive (Schwarzer et al., 2021a; Yu et al., 2022). ST-DIM (Anand et al., 2019), ATC (Stooke et al., 2021) and BVS-DIM (Mengistu et al., 2022) incorporate temporal information in their contrastive objective, adapting similar techniques from the unsupervised video representation learning (Sermanet et al., 2018). Proto-RL (Yarats et al., 2021a) uses a SwAV-like objective to learn representation as well as guide effective exploration during pre-training. Similarly, CRL (Du et al., 2021) trains a policy to optimize a SimCLR loss, then shows transfer to RL, imitation learning and image classification. Closer to our approach, SGI (Schwarzer et al., 2021b) pretrains both an encoder and forward prediction model by minimizing the distance between predictions and target latents using BYOL, and the encoder is recycled during RL for improved data efficiency. While different in spirit, many model based methods also train an encoder from a corpus of trajectory, either by explicit pixel reconstruction Kaiser et al. (2020); Hafner et al. (2021) or in embedding space Ye et al. (2021); Schrittwieser et al. (2020). Self-supervised representations have also been used for imitation learning (Aytar et al., 2018; Pari et al., 2021) as well as exploration (Burda et al., 2019a).

## 2.2 REPRESENTATION PROBING IN REINFORCEMENT LEARNING

Some prior work (Racah and Pal, 2019; Guo et al., 2018; Anand et al., 2019; Higgins et al., 2018; Dittadi et al., 2022) evaluate the quality of their pretrained representations by probing for ground truth state variables such as agent/object locations, game scores or model-specific quantities (eg. ELBO). Das et al. (2020) propose to probe representations with natural language question-answering. Despite the efficiency of these probing methods, their designs are highly domain-specific and require careful handcrafting for each environment. In addition, they fail to demonstrate the actual correlation between probing and RL performances, which makes their practical usefulness uncertain. On the other hand, the authors of ATC (Stooke et al., 2021) propose to evaluate representations by finetuning for RL tasks using the pretrained encoder with weights frozen. Similarly, Laskin et al. (2021) propose a unified benchmark for SSL methods in continuous control but still require full RL training. Our work seeks to bridge these two approaches by demonstrating the correlation between linear probing and RL performances, as well as designing probing tasks that are generalizable across environments.

## 3 A FRAMEWORK TO DEVELOP UNSUPERVISED REPRESENTATIONS FOR RL

In this section, we detail our proposed framework for training and evaluating unsupervised representations for reinforcement learning.

### 3.1 UNSUPERVISED PRE-TRAINING

The network is first pre-trained on a large corpus of trajectories. Formally, we define a trajectory $\mathcal{T}_i$ of length $T_i$ as a sequence of tuples $\mathcal{T}_i = [(o_t, a_t) \mid t \in [1, T_i]]$, where $o_t$ is the observation of the state at time $t$ in the environment and $a_t$ was the action taken in this state. This setting is closely related to Batch RL (Lange et al., 2012), with the crucial difference that the reward is not being observed. In particular, it should be possible to use the learned representations to maximize *any* reward (Touati and Ollivier, 2021). The training corpus corresponds to a set of such trajectories: $\mathcal{D}_{\text{unsup}} \{\mathcal{T}_1, \cdots, \mathcal{T}_n\}$. We note that the policy used to generate this data is left unspecified in this formulation, and is bound to be environment-specific. Since unsupervised methods usually necessitate a lot of data, this pre-training corpus is required to be substantial. In some domains, it might be straightforward to collect a large number of random trajectories to constitute $\mathcal{D}_{\text{unsup}}$. In some other cases, like self-driving, where generating random trajectories is undesirable, expert trajectories from humans can be used instead.

The goal of the pre-training step is to learn the parameters $\theta$ of an encoder $\text{ENC}_\theta$ which maps any observation $o$ of the state $s$ (for example raw pixels) to a representation $e = \text{ENC}_\theta(o)$. This representation must be amenable for the downstream control task, for example learning a policy.

### 3.2 EVALUATION

In general, the evaluation of RL algorithms is tricky due to the high variance in performance (Henderson et al., 2018). This requires evaluating many random seeds, which creates a computational burden. We side-step this issue by formulating an evaluation protocol which is light-weight and purely

supervised. Specifically, we identify two proxy supervised tasks that are broadly applicable and relevant for control. We further show in the experiment section that they are *sound*, in the sense that models' performance on the proxy tasks strongly correlates with their performance in the downstream control task of interest. Similar to the evaluation protocol typically used for computer vision models, we rely on *linear probing*, meaning that we train only a linear layer on top of the representations, which are kept frozen.

**Reward Probing**  Our first task consists in predicting the reward observed in a given state. For this task, we require a corpus of trajectories $\mathcal{D}_{\text{rew}} = \{\mathcal{T}'_1, \cdots, \mathcal{T}'_m\}$ for which the observed rewards are known, i.e. $\mathcal{T}'_i = [(o_t, a_t, r_t) \mid t \in [1, T_i]]$

In the most general setting, it can be formulated as a regression problem, where the goal is to minimize the following loss:

$$\mathcal{L}(\psi)_{\text{reward-reg}} = \frac{1}{|\mathcal{D}_{\text{rew}}|} \sum_{\mathcal{T}'_i \in \mathcal{D}_{\text{rew}}} \frac{1}{|\mathcal{T}'_i|} \sum_{(o_t, a_t, r_t \in \mathcal{T}'_i)} \|l_\psi(\text{ENC}_\theta(o_t)) - r_t\|_2$$

Here, the only learnt parameters $\psi$ are those of the linear prediction layer $l_\psi$.

In practice, in many environments where rewards are sparse, the presence or absence of a reward is more important than its magnitude. To simplify the problem in those cases, we can cast it as a binary prediction problem instead (this could be extended to ternary classification if the sign of the reward is of interest):

$$\mathcal{L}(\psi)_{\text{reward-classif}} = \frac{1}{|\mathcal{D}_{\text{rew}}|} \sum_{\mathcal{T}'_i \in \mathcal{D}_{\text{rew}}} \frac{1}{|\mathcal{T}'_i|} \sum_{(o_t, a_t, r_t \in \mathcal{T}'_i)} \text{BinaryCE}(\mathbb{1}_{\mathbb{R}_{>0}}(r_t), l_\psi(\text{ENC}_\theta(o_t)))$$

Reward prediction is closely related to value prediction, a central objective in RL that is essential for value-based control and the critic in actor-critic methods. The ability to predict instantaneous reward, akin to predicting value with a very small discount factor, can be viewed as a lower bound on the learned representation's ability to encode the value function, and has been demonstrably helpful for control, particularly in sparse reward tasks (Jaderberg et al., 2017). Thus, we hypothesize reward prediction accuracy to be a good probing proxy task for our setting as well.

**Action prediction**  Our second task consists in predicting the action taken by an expert in a given state. For this task, we require a corpus of trajectories $\mathcal{D}_{\text{exp}} = \{\mathcal{T}_1, \cdots, \mathcal{T}_n\}$ generated by an expert policy. We stress that this dataset may be much smaller than the pretraining corpus since we only require to fit and evaluate a linear model. The corresponding objective is as follows:

$$\mathcal{L}(\psi)_{\text{action-classif}} = \frac{1}{|\mathcal{D}_{\text{exp}}|} \sum_{\mathcal{T}_i \in \mathcal{D}_{\text{exp}}} \frac{1}{|\mathcal{T}_i|} \sum_{(o_t, a_t \in \mathcal{T}'_i)} \text{CrossEntropy}(a_t, l_\psi(\text{ENC}_\theta(o_t)))$$

This task is closely related to imitation learning, however, we are not concerned with the performance of the policy that we learn as a by-product.

## 4   SELF PREDICTIVE REPRESENTATION LEARNING FOR RL

In our work, we focus on evaluating and improving a particular class of unsupervised pretraining algorithms that involves using a transition model to predict its own representations in the future (Schwarzer et al., 2021b; Guo et al., 2018; Gelada et al., 2019). This pretraining modality is especially well suited for RL, since the transition model can be conditioned on agent actions, and can be repurposed for model-based RL after pretraining. Our framework is depicted in Fig.2. In this section, we present the main design choices, and we investigate their performance in Section 5.

### 4.1   TRANSITION MODELS

Our baseline transition model is a 2D convolutional network applied directly to the spatial output of the convolutional encoder (Schwarzer et al., 2021b; Schrittwieser et al., 2020). The network consists

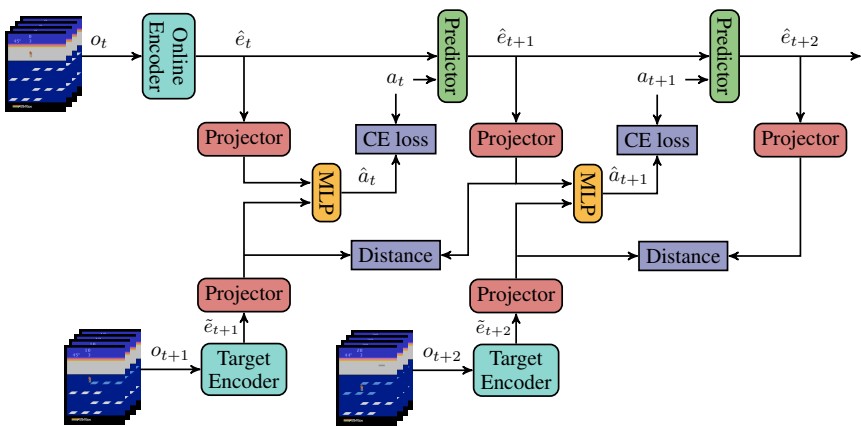

Figure 2: Model diagram. The observations consist of a stack of 4 frames, to which we apply data augmentation before passing them to a convolutional encoder. The predictor is a recurrent model outputting future state embeddings given the action. We supervise with an inverse modeling loss (cross entropy loss on the predicted transition action) and an SSL loss (distance between embeddings).

of two 64-channel convolutional layers with 3x3 filters. The action is represented as a one-hot encoding spatially replicated (in a 2D map) and concatenated with the representation input along the channel dimension.

We believe a well-established sequence modeling architecture such as GRU can serve as a superior transition model. Its gating mechanisms should be better at retaining information from both the immediate and distant past, especially helpful for learning dynamics in a partially observable environment.

$$
\begin{aligned}
\text{Encoder}: \quad & \hat{e_0} = e_0 = \text{ENC}_\theta(o_0) \\
\text{RecurrentModel}: \quad & \hat{e}_t = f_\phi(\hat{e}_{t-1}, a_{t-1})
\end{aligned}
$$

In addition to the deterministic GRU model above, we also experiment with a GRU variant where we introduce stochastic states to allow our model to generalize better to stochastic environments, such as Atari with sticky actions (Machado et al., 2018). Our model is based on the RSSM from DreamerV2 (Hafner et al., 2021), with the main difference being that while pixel reconstruction is used as the SSL objective in the original work, we minimize the distance between predictions and targets purely in the latent space. Following DreamerV2, we optimize the latent variables using straight-through gradients (Bengio et al., 2013), and minimize the distance between posterior ($z$) and prior ($\hat{z}$) distributions using KL loss.

$$
\begin{aligned}
\text{Encoder}: \quad & e_t = \text{ENC}_\theta(o_t) \\
\text{RecurrentModel}: \quad & h_t = f_\phi(h_{t-1}, z_{t-1}, a_{t-1}) \\
\text{PosteriorModel}: \quad & z_t \sim p_\phi(z_t | h_t, e_t) \\
\text{PriorPredictor}: \quad & \hat{z}_t \sim j_\phi(\hat{z}_t | h_t) \\
\text{LatentMerger}: \quad & \hat{e}_t = g_\phi(h_t, z_t)
\end{aligned}
$$

## 4.2 PREDICTION OBJECTIVES

The objective of self predictive representation learning is to minimize the distance between the predicted and the target representations, while ensuring that they do not collapse to a trivial solution. Our baseline prediction objective is BYOL (Grill et al., 2020), which is used in SGI (Schwarzer et al., 2021b). The predicted representation $\hat{e}_{t+k}$, and the target representation $\tilde{e}_{t+k}$ are first projected to lower dimensions to produce $\hat{y}_{t+k}$ and $\tilde{y}_{t+k}$. BYOL then maximizes the cosine similarity between the predicted and target projections, using a linear prediction function $q$ to translate from $\hat{y}$ to $\tilde{y}$:

$$L_\theta^{BYOL}(\hat{y}_{t:t+k}, \tilde{y}_{t:t+k}) = -\sum_{k=1}^{K} \frac{q(\hat{y}_{t+k}) \cdot \tilde{y}_{t+k}}{\|q(\hat{y}_{t+k})\|_2 \cdot \|\tilde{y}_{t+k}\|_2}$$

In the case of BYOL, the target encoder and projection module are the exponentially moving average of the online weights, and the gradients are blocked on the target branch.

As an alternative prediction objective, we experiment with Barlow Twins (Zbontar et al., 2021). Similar to BYOL, Barlow Twins minimizes the distance of the latent representations between the online and target branches; however, instead of using a predictor module and stop gradient on the target branch, Barlow Twins avoids collapse by pushing the cross-correlation matrix between the projection outputs on the two branches to be as close to the identity matrix as possible. To adapt Barlow Twins, we calculate the cross correlation across batch and time dimensions:

$$L^{BT}(\hat{y}_{t:t+k}, \tilde{y}_{t:t+k}) = \sum_i (1 - C_{ii})^2 + \lambda \sum_{i,j \neq i} C_{ij}^2 \text{ where } C_{ij} = \frac{\sum_{b,t}(\hat{y}_{b,t,i}) \cdot (\tilde{y}_{b,t,j})}{\sqrt{\sum_{b,t}(\hat{y}_{b,t,i})^2} \cdot \sqrt{\sum_{b,t}(\tilde{y}_{b,t,j})^2}}$$

where $\lambda$ is a positive constant trading off the importance of the invariance and covariance terms of the loss, $C$ is the cross-correlation matrix computed between the projection outputs of two branches along the batch and time dimensions, $b$ indexes batch samples, $t$ indexes time, and $i, j$ index the vector dimension of the projection output.

By enabling gradients on both the prediction and target branches, the Barlow objective pushes the predictions towards the representations, while regularizing the representations toward the predictions. In practice, learning the transition model takes time and we want to avoid regularizing the representations towards poorly trained predictions. To address this, we apply a higher learning rate to the prediction branch. We call this technique Barlow Balancing, and implement it in Algorithm 1.

---

**Algorithm 1:** PyTorch-style pseudocode for Barlow Balancing

$$\text{BarlowLoss} = \mu * L^{BT}(\hat{y}, \tilde{y}.\text{detach}()) + (1 - \mu) * L^{BT}(\hat{y}.\text{detach}(), \tilde{y})$$

---

### 4.3 OTHER SSL OBJECTIVES

SGI's authors (Schwarzer et al., 2021b) showed that in the absence of other SSL objectives, pretraining with BYOL prediction objective alone results in representation collapse; the addition of inverse dynamics modeling loss is necessary to prevent collapse, while the addition of goal-oriented RL loss results in minor downstream RL performance improvement. In inverse dynamics modeling, the model is trained using cross-entropy to model $p(a_t|\hat{y}_{t+k}, \tilde{y}_{t+k+1})$, effectively predicting the transition action between two adjacent states. The goal-oriented loss tries to predict distance to states in the near future from the sampled trajectories (details in Appendix).

## 5 RESULTS

### 5.1 EXPERIMENTAL DETAILS

We conduct experiments on the Arcade Learning Environment benchmark (Bellemare et al., 2013). Given the multitude of pretraining setups we investigate, we limit our experiment to 9 Atari games[1].

**Pretraining** We use the publicly-available DQN replay dataset (Agarwal et al., 2020), which contains data from training a DQN agent for 50M steps with sticky action (Machado et al., 2018). We select 1.5 million frames from the 3.5 to 5 millionth steps of the replay dataset, which constitutes trajectories of a weak, partially trained agent. We largely follow the recipe of SGI (Schwarzer et al., 2021b), where we jointly optimize the self prediction, goal-conditioned RL, and inverse dynamics modeling

---

[1]Amidar, Assault, Asterix, Boxing, Demon Attack, Frostbite, Gopher, Krull, Seaquest

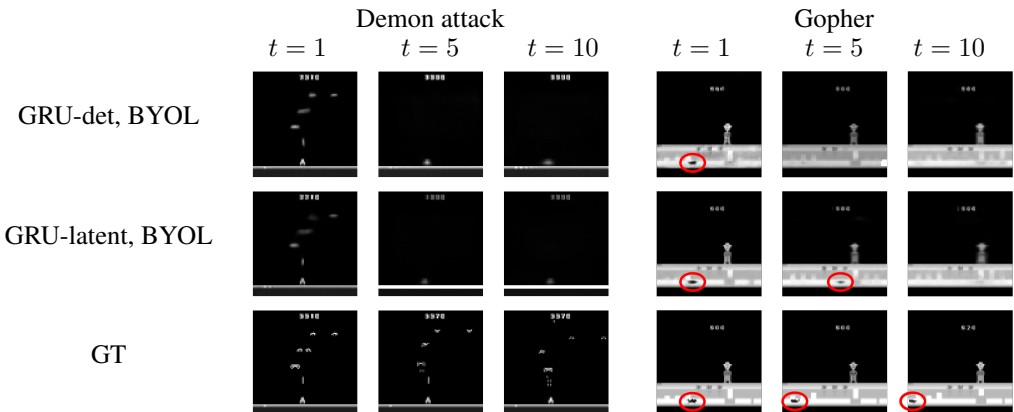

Figure 3: Decoding results, using a de-convolutional model to predict the pixel values from frozen state representations. Both games exhibit stochastic behaviours. In Demon attack, both models fail to capture the position of the enemies. In Gopher, the enemy (circled in red) is moving randomly, but thanks to the latent variable, the GRU-latent model is able to predict a possible position, while the deterministic model regresses to the mean.

losses for 20 epochs; in some of our experiments we remove one or both of the last two objectives. We use the data-augmentations introduced by Yarats et al. (2021b). All experiments are performed on a single MI50 AMD GPU, and the pretraining process took 2 to 8 days depending on the model.

**Reward probing** We focus on the simplified binary classification task of whether a reward occurs in a given state. We use 100k frames from the 1-1.1 millionth step of the replay dataset, with a 4:1 train/eval split. We train a logistic regression model on frozen features using the Cyanure (Mairal, 2019) library, with the MISO algorithm (Mairal, 2015) coupled with QNING acceleration (Lin et al., 2019) for a maximum of 300 steps. We do not use any data augmentation. We report the mean F1 averaged across all 9 games. On a MI50 AMD GPU, each probing run takes 10 minutes.

**Action probing** We use the last 100k (4:1 train/eval split) frames of the DQN replay dataset, which correspond to a fully trained DQN agent. We train a linear layer on top of frozen, un-augmented features for 12 epochs with softmax focal loss (Lin et al., 2017) using SGD optimizer with learning rate 0.2, batch size 256, 1e-6 weight decay, stepwise scheduler with step size 10 and gamma 0.1. We report the Multiclass F1 (weighted average of F1 scores of each class) averaged across all games.

**RL evaluation** We focus on the Atari 100k benchmark (Kaiser et al., 2020), where only 100k interactive steps are allowed by the agent. This is roughly equivalent to two hours of human play, providing an approximation for human level sample-efficiency. We follow Schwarzer et al. (2021b) training protocol using the Rainbow algorithm (Hessel et al., 2018) with the following differences: we freeze the pretrained encoder (thus only training the Q head), do not apply auxiliary SSL losses while fine-tuning, and finally disable noisy layers and rely instead on $\epsilon$-greedy exploration. This changes are made to make the RL results reflect as closely as possible the performance induced by the quality of the representations. On a MI50 AMD GPU, each run takes between 8 and 12 hours.

We evaluate the agent's performance using human-normalized score (HNS), defined as $(agentscore - randomscore)/(humanscore - randomscore)$. We calculate this per game, per seed by averaging scores over 100 evaluation trajectories at the end of training. For aggregate metrics across games and seeds, we report the median and interquartile mean (IQM). For median, we first average the HNS across seeds for each game, and report the median of the averaged HNS values. For IQM, we first take the middle 50% of scores across both seeds and games, then report the average. While median is commonly reported for Atari100k, recent work has recommended IQM as a superior aggregate metric for the RL setting due to its smaller uncertainty (Agarwal et al., 2021); we also follow the cited work to report the 95% bootstrapped confidence intervals for these aggregate metrics.

Unless specified otherwise, the experiments use the medium ResNet-M from Schwarzer et al. (2021b), and the inverse dynamics loss as an auxiliary loss. In BYOL experiments, the target network is an exponential moving average of the online network, while in Barlow Twins both networks are identical, following the original papers. For additional details regarding model architectures and hyperparameters used during pretraining and RL evaluation, please refer to Appendix.

## 5.2 IMPACT OF TRANSITION MODELS AND PREDICTION OBJECTIVES

Table 1: F1 scores on probing tasks for different transition models and prediction objectives. All standard deviations are on the order of 1e-4

| Pred Obj | Transition | Reward | Action |
|---|---|---|---|
| BYOL | Conv-det | 64.9 | 22.7 |
| | GRU-det | 62.2 | 26.8 |
| | GRU-latent | 63.4 | 23.2 |
| $Barlow_{0.7}$ | Conv-det | 52.7 | 24.9 |
| | GRU-latent | 67.5 | 26.2 |

Table 2: F1 scores on probing tasks for different Barlow variants. All standard deviations are on the order of 1e-4 which we omit below.

| Pred Obj | Reward | Action |
|---|---|---|
| $Barlow_{0.5}$ | 65.0 | 26.3 |
| $Barlow_{0.7}$ | 67.5 | 26.2 |
| $Barlow_1$ | 65.0 | 24.7 |
| $Barlow_{rand}$ | 67.7 | 25.8 |

In table 1, we report the mean probing F1 scores for the convolutional, deterministic GRU, and latent GRU transition models trained using either the BYOL or Barlow prediction objective. When using the BYOL objective, the relative probing strengths for the different transition models are somewhat ambiguous: while the convolutional model results in better reward probing F1, the GRU models are superior in terms of expert action probing.

Interestingly, we observe that after replacing BYOL with Barlow, the probing scores for the latent model improve, while those of the deterministic models deteriorate. Overall, the particular combination of pre-training using the GRU-latent transition model with the Barlow prediction objective results in representations with the best overall probing qualities. Since the deterministic model's predictions are likely to regress to the mean, allowing gradients to flow through the target branch in the case of Barlow objective can regularize the representations towards poor predictions, and can explain their inferior probing performance. Introducing latent variables can alleviate this issue through better predictions.

We stress that the transition models are not used during probing, only the encoder is. These experiments show that having a more expressive forward model during the pre-training has a direct impact on the quality of the learnt representations. In Fig.3, we investigate the impact of the latent variable on the information contained in the representations, by training a decoder on frozen features.

In table 2, we show the results from experimenting with different variants of the Barlow objective. We find that using a higher learning rate for the prediction branch ($Barlow_{0.7}$, with 7:3 prediction to target lr ratio) results in better probing outcome than using equal learning rates ($Barlow_{0.5}$) or not letting gradients flow in the target branch altogether ($Barlow_1$, here the target encoder is a copy of the online encoder). This suggests that while it is helpful to regularize the representations towards the predictions, there is a potential for them being regularized towards poorly trained ones. This can be addressed by applying a higher learning rate on the prediction branch.

We also demonstrate that using a frozen, random target network ($Barlow_{rand}$) results in good features, and in our experiments it gets the best reward probing performance. This contradicts findings from the vision domain (Grill et al., 2020), but corroborates self-supervised results from other domains such as speech (Chiu et al., 2022). Random networks have also been shown to exhibit useful inductive biases

Table 3: Representation probing and RL results. Mean binary F1 for reward, mean multiclass F1 for next action (all stdev ≈ 1e-4). RL metrics are aggregated on 10 seeds of 9 games. The 95% CIs are estimated using the percentile bootstrap with stratified sampling (Agarwal et al., 2021).

| Backbone | Transition | Objectives | Reward | Action |
|---|---|---|---|---|
| L | GRU-lat | $Barlow_{rand}$, inv | 70.3 | 26.7 |
| L | GRU-lat | $Barlow_{0.7}$, inv | 69.0 | 27.7 |
| M | GRU-lat | $Barlow_{rand}$, inv | 67.7 | 25.8 |
| M | GRU-lat | $Barlow_{0.7}$, inv | 67.4 | 26.2 |
| M | GRU-lat | BYOL, goal, inv | 63.4 | 23.2 |
| M | GRU-det | BYOL, goal, inv | 62.2 | 26.9 |
| M | Conv-det | BYOL, goal, inv | 64.9 | 22.7 |
| M | GRU-lat | $Barlow_{0.7}$ | 56.2 | 24.4 |
| M | Conv-det | $Barlow_{0.7}$, goal, inv | 52.7 | 24.8 |
| S | None | ATC Stooke et al. (2021) | 51.3 | 22.9 |
| S | None | ST-DIM Anand et al. (2019) | 34.6 | 21.3 |
| S | None | VAE-T Stooke et al. (2021) | 33.2 | 20.4 |

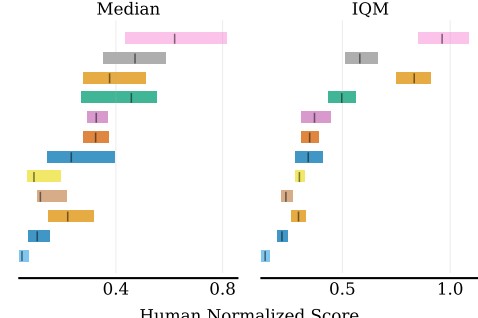

for exploration (Burda et al., 2019b;a). An explanation is that random targets act as a regularization that prevent partial collapse by enforcing a wide range of features to be encoded by the model.

## 5.3 IMPACT OF AUXILIARY SSL OBJECTIVES AND ENCODERS

Table 4: F1 scores on probing tasks for different auxiliary objectives. All stdev $\approx$ 1e-4.

| SSL Objs | Reward | Action |
|---|---|---|
| BYOL, inv, goal | 63.4 | 23.2 |
| BYOL, inv | 57.3 | 22.6 |
| BYOL | 25.9 | 5.9 |
| Barlow$_{0.7}$, inv, goal | 66.5 | 26.2 |
| Barlow$_{0.7}$, inv | 67.5 | 26.2 |
| Barlow$_{0.7}$ | 56.2 | 24.4 |

Table 5: F1 scores on probing tasks for different encoders. All stdev $\approx$ 1e-4.

| Pred Obj | Encoder | Reward | Action |
|---|---|---|---|
| Barlow$_{0.7}$ | Res-M | 67.5 | 26.2 |
| | Res-L | 69.0 | 27.7 |
| Barlow$_{rand}$ | Res-M | 67.7 | 25.8 |
| | Res-L | 70.3 | 26.7 |

**SSL objective** Although pretraining with multiple objectives can sometimes result in better downstream performance, in practice they also make it harder to tune for hyperparameters and debug, therefore it is desirable to use the least number of objectives that can result in comparable performance.

In table 4, we show the effects of inverse dynamics modeling (inv) and goal-conditioned RL (goal) objectives on probing performance. The BYOL model experiences partial collapse without the inverse dynamics modeling loss, while the addition of goal loss improves the probing performance slightly. This is in congruence with results reported by Schwarzer et al. (2021b) for the same ablations.

The Barlow-only model performs significantly better than the BYOL-only model in terms of probing scores, indicating that the Barlow objective is less prone to collapse in the predictive SSL setting. Similar to the BYOL model, the Barlow model can also be improved with inverse dynamics modeling, while the addition of goal loss has a slight negative impact.

**Encoders** SGI (Schwarzer et al., 2021b) showed that using bigger encoders during pretraining results in improved downstream RL performance. We revisit this topic from the point of finding out whether the pretrained representations from bigger networks also have better probing qualities. We experiment with the medium (ResNet-M) and large (ResNet-L) residual networks from SGI. In table 5 we show that Barlow models pretrained using the larger ResNet have improved probing scores.

## 5.4 CORRELATIONS BETWEEN PROBING AND RL PERFORMANCES

If our goal is to use linear probing as a guide to identify superior pretraining setup for RL, then they are only useful to the extent to which they correlate with the actual downstream RL performance. We perform RL evaluations for 9 representative setups (the best settings from each of table 1,2,4,5), as well as two contrastive methods: ST-DIM (Anand et al., 2019) and ATC (Stooke et al., 2021); and a reconstruction-based method VAE-T (Stooke et al., 2021)[2]. We report their probing and aggregate RL metrics in table 3, with the confidence intervals of the aggregate RL metrics depicted on the right. We find that the rank correlations between reward and action probing F1 scores and the RL aggregate metrics are significant (Figure 1). In summary, our results show the proposed probing scheme is a reliable guide for designing pretraining setups that deliver significant downstream RL performance improvements.

## 6 CONCLUSION

In this paper we have investigated the opportunity to replace costly RL evaluation with lightweight linear probing task to assess the quality of learned representations. Reward and action probing are task-agnostic and should cover most practical applications. Using this methodology to guide us, we have demonstrated the impact of a number of key design choices in the pre-training methodology. We hope that these results encourage the research community to systematically explore the design space to further improve the quality of self-supervised representations for RL.

---

[2]See appendix for details on ATC, ST-DIM and VAE-T

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

# A    Models and Hyper-parameters

## A.1    Backbones

**M and L models** are ResNet-M and ResNet-L from SGI (Schwarzer et al., 2021b). The ResNet-M encoder consists of inverted residual blocked with an expansion ratio of 2, with batch normalization applied after each convolutional layer; it uses 3 groups with 32, 64, and 64 channels, and has 3 residual blocks per group; it down-scales the input by a factor of 3 in the first group and 2 in the latter 2 groups. This yields a representation of shape 64x7x7 when applied to 84x84-dimensional Atari frames. ResNet-L uses 3 groups with 48, 96, and 96 channels, and has 5 residual blocks per group; it uses a larger expansion ratio of 4, producing a representation shape of 96x7x7 from an 84x84 frame. This enlargement increases the number of parameters by approximately a factor of 5.

**S model** is the model used in Stooke et al. (2021). It consists of three convolutional layers, with $[32, 64, 64]$ channels , kernel sizes $[8, 4, 3]$, and strides $[4, 2, 1]$, listed from first to last layer.

## A.2    Transition Models

We experimented with three transition models: convolutional model, deterministic GRU, and latent GRU. Our convolutional model is based on SGI (Schwarzer et al., 2021b). The input into the convolutional transition model is the concatenation of the spatially replicated 2D action map and the representation $e_t$ along the channel dimension. The network itself consists of two 64-channel convolutional layers with 3x3 filters, separated by ReLU activation and batch normalization layers.

The deterministic GRU has hidden dimension 600 and input dimension 250. The input $a_t$ is prepared by passing the one-hot action vector through a 250 dimensional embedding layer. The initial hidden state $\hat{e}_0$ is generated by projecting the representation $e_0$ through a 600 dimensional linear layer with ELU activation and dropout. Layer normalization is applied to the hidden input at all timesteps.

The latent GRU model is based on Dreamerv2's RSSM (Hafner et al., 2021), and is consisted of a recurrent model, posterior model, prior predictor, and latent merger. The recurrent model has a hidden dimension and input dimension of 600. The initial hidden state $h_0$ and input $z_0$ are zero vectors. The flattened stochastic variables $z_t$ and one-hot action vector $a_t$ are first concatenated and then projected to 600 dimension through a linear layer with ELU activation, before being passed into the recurrent model as input. Layer normalization is applied to the hidden input at all non-zero timesteps.

The posterior model is a two-layer MLP with 600 dimensional bottleneck separated by ELU activation. It takes the concatenation of representation $e_t$ and recurrent hidden output $h_t$ as input, and outputs a 1024 dimensional vector representing the 32 dimensional logits for 32 latent categorical variables. $z_t$ is sampled from the posterior logits. The prior model is a two-layer MLP with 600 dimensional bottleneck separated by ELU activation. Its output format is same as that of the posterior model. $\hat{z}_t$ is sampled from the prior logits. The latent merger is a linear layer that projects the concatenation of $h_t$ and flattened $z_t$ to the same dimension of representation $e_t$.

## A.3    SSL Projection Module

In the case of the deterministic GRU, $\hat{e}$ is first projected to the same dimension of representation through a linear layer. Henceforth we shall assume that $\hat{e}$ underwent this step for $\text{GRU}_{\text{det}}$.

The predicted representation $\hat{e}$ and target representation $\tilde{e}$ are projected to 1024 dimensional vectors $\hat{y}$ and $\tilde{y}$ through a linear layer. The BYOL objective involves processing $\hat{y}$ with an additional linear layer $q$ with output dimension 1024. The Barlow objective involves applying batch normalization to $\hat{y}$ and $\tilde{y}$ prior to taking the covariance and variance losses.

The inverse dynamics model is a two-layer MLP with 256 dimensional bottleneck separated by ReLU activation. It takes the concatenation of $\hat{y}_t$ and $\tilde{y}_{t+1}$ as input, and outputs logits with dimension equivalent to number of actions.

## A.4 ATC, VAE-T, ST-DIM

We use the implementation, hyperparameters and architecture from the codebase of (Stooke et al., 2021) and (Stooke and Abbeel, 2019) for these models. We change the dataset to the one used in all our experiments We use the dataset described in section 5 to train these models, and train all methods for 58,500 updates. ATC (Augmented-Temporal Contrast) uses InfoNCE loss between output of the momentum encoder and online branch applied to different augmentations of an image to pre-train the encoder. VAE-T from Stooke et al. (2021) uses variational auto-encoder (Kingma and Welling, 2014) objective to reconstruct the frame from the next time step given an image at the current time step. ST-DIM (Anand et al., 2019) also uses InfoNCE objective, and in additional to traditional global-global infomax, introduces global-local infomax by using local representations taken from the feature map output of the convolutional encoder and the global pooled feature vector as positive pairs. For more details, we refer the reader to the referenced works.

## A.5 IMAGE RECONSTRUCTION MODEL

We used a decoder architecture that mirrors the structure of the ResNet-M encoder. In decoding, instead of transposed convolutions we used upsampling with the nearest value followed by a regular convolution (Odena et al., 2016). We used mean squared error between the reconstructed pixels and the target image as the training criterion. Models were trained and evaluated on the same data as reward and action probing, for 30 epochs using Adam optimizer with learning rate 0.001.

## A.6 HYPERPARAMETERS

See tables 6, 7, 8, 9 for hyperparameter values. For ATC, ST-DIM and VAE-T hyperparameters, see Stooke et al. (2021).

## A.7 IMAGE AUGMENTATION

We use the same image augmentations as used in SGI (Schwarzer et al., 2021b), which itself used the augmentations in DrQ (Yarats et al., 2021b), in both pretraining and fine-tuning. We specifically apply random crops (4 pixel padding and 84x84 crops) and image intensity jittering.

## A.8 GOAL-ORIENTED RL LOSS

Goal-oriented RL loss is taken directly from SGI (Schwarzer et al., 2021b). This objective trains a goal-conditional DQN, with rewards specified by proximity to sampled goals. First, a goal $g$ is sampled to be the state encoding either of the near future in the current trajectory (up to 50 steps in the future), or, with probability of 20%, of the future state in another trajectory in the current batch. Then, we add Gaussian noise to obtain the final goal $g$: $g \leftarrow \alpha n + (1 - \alpha)g$, where $\alpha \sim \text{Uniform}(0.5)$, and $n$ is a vector sampled from isotropic Gaussian normalized to have length of 1. Then, in order to obtain the reward of taking action $a_t$ going from state $s_t$ to $s_{t+1}$, we first encode the states with the target encoder $\tilde{e}_t = \text{ENC}_{\text{target}}(o_t)$, $\tilde{e} + 1 = \text{ENC}_{\text{target}}(o_{t+1})$. Then, we calculate the reward as:

$R(\tilde{e}_t, \tilde{e}_{t+1}) = d(\tilde{e}_t, g) - d(\tilde{e}_{t+1}, g)$, where $d(\tilde{e}_t, g) = \exp\left(2\frac{\tilde{e}_t \cdot g}{\|\tilde{e}_t\|_2 \cdot \|g\|_2} - 2\right)$. We use FiLM (Perez et al., 2018) to condition the Q-function $Q(o_t, a_t, g)$ on $g$, and optimize the model using DQN (Mnih et al., 2015).

## B FORWARD MODEL PROBING

While our principal goal is to demonstrate the correlation between representation probing and offline RL performances, we also apply the reward probing technique to predictions in order to evaluate the qualities of transition models under different pretraining setups.

In table 10, we show the effects of using different transition models during pretraining on prediction probing performance. All models are trained with ResNet-M encoder and inverse loss. Goal loss is also applied to the BYOL models.

Table 6: Hyperparameters for pretraining and RL evaluation.

| | Parameter | Setting |
|---|---|---|
| Pretrain & RL | Gray-scaling | True |
| | Observation down-sampling | 84x84 |
| | Frames stacked | 4 |
| | Action repetitions | 4 |
| | Sticky Action | True |
| | Reward clipping | [-1, 1] |
| | Terminal on loss of life | True |
| | Optimizer | Adam |
| | Optimizer: learning rate | 0.0001 |
| | Optimizer: $\beta_1$ | 0.9 |
| | Optimizer: $\beta_2$ | 0.999 |
| | Optimizer: $\epsilon$ | 0.00015 |
| | Minibatch Size | 64 |
| | Max gradient norm | 10 |
| Pretrain | Prediction Depth | 10 |
| | Epochs | 20 |
| | Goal loss weight | 0 or 1 |
| | Inverse loss weight | 0 or 1 |
| RL | Max frams per episode | 108K |
| | Update | Distributional Q |
| | Dueling | True |
| | Support of Q-distribution | 51 |
| | Discount factor | 0.99 |
| | Priority exponent | 0.5 |
| | Priority correction | $0.4 \to 1$ |
| | Exploration | $\epsilon$-greedy |
| | Training steps | 100K |
| | Evaluation trajectories | 100 |
| | Min replay size for sampling | 2000 |
| | Replay period every | 1 step |
| | Updates per step | 2 |
| | Multi-step return length | 10 |
| | Q network: channels | 32,64,64 |
| | Q network: filter size | 8x8, 4x4, 3x3 |
| | Q network: stride | 4,2,1 |
| | Q network: hidden units | 256 |
| | Non-linearity | ReLU |
| | Target network: update period | 1 |

Table 7: SSL specific hyperparameters.

| | Parameter | Setting |
|---|---|---|
| BYOL | loss weight | 1 |
| | $\tau$ | 0.99 |
| Barlow | loss weight | 0.002 |
| | $\lambda$ | 0.0051 |

Table 8: GRU-latent specific hyperparameters.

| Parameter | Setting |
|---|---|
| kl loss weight | 0.1 |
| kl balance | 0.95 |

Table 9: Optimal RL learning rate for different setups. Identified by sweeping through $2.5e{-}5$, $5e{-}5$, $1e{-}4$, $2e{-}4$ and evaluated on games frostbite, assault, gopher, and demon attack.

| Encoder | Transition | Objectives | Learning Rate |
|---|---|---|---|
| ResNet-M | Conv-det | BYOL, goal, inv | $2e{-}4$ |
| ResNet-M | GRU-det | BYOL, goal, inv | $2e{-}4$ |
| ResNet-M | GRU-latent | BYOL, goal, inv | $2e{-}4$ |
| ResNet-M | GRU-latent | Barlow$_{0.7}$, inv | $1e{-}4$ |
| ResNet-M | GRU-latent | Barlow$_{rand}$, inv | $5e{-}5$ |
| ResNet-L | GRU-latent | Barlow$_{0.7}$, inv | $5e{-}5$ |
| ResNet-L | GRU-latent | Barlow$_{rand}$, inv | $1e{-}4$ |

Table 10: Mean reward probing F1 scores for pretraining setups with different transition models. Evaluated on $5^{th}$ and $10^{th}$ predictions. All standard deviations are on order of 1e-4.

| Pred Obj | Transition | Pred 5 | Pred 10 |
|---|---|---|---|
| BYOL | Conv-det | 33.1 | 28.4 |
| | GRU-det | 33.0 | 27.4 |
| | GRU-latent | 33.4 | 28.9 |
| $Barlow_{0.7}$ | Conv-det | 32.0 | 27.6 |
| | GRU-det | 30.1 | 25.0 |
| | GRU-latent | 39.5 | 30.2 |

Table 11: Mean reward probing F1 scores for pretraining setups with different prediction objectives. Evaluated on $5^{th}$ and $10^{th}$ predictions. All standard deviations are on order of 1e-4.

| Pred Obj | Pred 5 | Pred 10 |
|---|---|---|
| BYOL | 33.4 | 28.9 |
| $Barlow_{0.5}$ | 40.2 | 30.2 |
| $Barlow_{0.7}$ | 39.5 | 30.2 |
| $Barlow_1$ | 37.4 | 29.7 |
| $Barlow_{rand}$ | 36.8 | 27.5 |

In the deterministic setting, the predictions of the GRU model are worse than those of the convolutional model. The introduction of stochasticity appears to fix the underlying issue for predictions, resulting in the latent GRU model having the best overall prediction probing performance.

One possible explanation for Conv-det having better predictions than GRU-det is that the spatial inductive bias in the convolutional kernels acts as a constraint and helps regularize the predictions from regressing to the mean. However, this is more effectively solved by the introduction of latent variables into GRU during training and inference.

In table 11, we show the effects of using different prediction objectives during pretraining on prediction probing performance. All models are trained with ResNet-M encoder, GRU-latent transition model, and inverse loss; goal loss is also applied to the BYOL model.

Comparing to the BYOL model, Barlow models generally have higher probing scores for predictions. We also note that for Barlow models, regularizing the representations towards the predictions (by setting Barlow Balance < 1) improves the qualities of predictions. This is likely because it makes the prediction task easier, making it more likely to learn a capable transition model.

This reasoning can also explain why the Barlow model with frozen, random target network achieves superior probing result for representation (table 2) but worse result for predictions compared to the other Barlow versions. Predicting a random target representation is likely more difficult than predicting a learned representation, and this may in turn encourage the model to rely more on learning a powerful encoder and posterior model, and less on learning an accurate transition model.

## C  FULL RL RESULTS

Table 12: Full RL Results for representative pretraining setups. Setup names are represented as *{encoder}-{transition model}-{ssl losses}*. **M** and **L** refer to ResNet M and ResNet L, **CD** is convolutional model, **GD** is deterministic GRU, **GL** is latent GRU, **By** and **Bt** refer to Byol and Barlow, **G** and **I** refer to goal and inverse losses.

| | Amidar | Assault | Asterix | Boxing | DemonAtt | Frostbite | Gopher | Seaquest | Krull |
|---|---|---|---|---|---|---|---|---|---|
| Random | 5.8 | 222.4 | 210.0 | 0.1 | 152.1 | 65.2 | 257.6 | 68.4 | 1598.0 |
| Human | 1719.5 | 742.0 | 8503.3 | 12.1 | 1971.0 | 4334.7 | 2412.5 | 42054.7 | 2665.5 |
| VAE-T | 56.1 | 526.8 | 355.3 | 28.1 | 830.8 | 394.0 | 484.2 | 311.9 | 3907.3 |
| ST-DIM | 64.7 | 492.2 | 335.0 | 19.8 | 524.5 | 269.7 | 354.2 | 376.1 | 2650.6 |
| ATC | 100.8 | 653.8 | 376.9 | 21.5 | 812.3 | 1002.1 | 601.5 | 456.4 | 3611.8 |
| M-CD-ByGI | 169.6 | 693.1 | 393.1 | 54.5 | 458.1 | 1058.9 | 1323.4 | 461.7 | 5541.4 |
| $M-CD-Bt_{0.7}GI$ | 206.1 | 545.6 | 500.0 | 21.5 | 357.7 | 518.5 | 880.5 | 482.4 | 4216.0 |
| M-GD-ByGI | 204.5 | 552.6 | 625.2 | 51.0 | 723.5 | 979.7 | 1299.2 | 597.1 | 5006.3 |
| M-GL-ByGI | 170.9 | 392.0 | 527.2 | 49.1 | 1842.9 | 541.9 | 1489.7 | 609.9 | 4753.9 |
| $M-GL-Bt_{0.7}$ | 97.9 | 846.0 | 442.5 | 53.9 | 311.5 | 461.5 | 731.0 | 622.1 | 4176.4 |
| $M-GL-Bt_{0.7}I$ | 189.9 | 861.8 | 426.4 | 63.2 | 1048.8 | 2020.1 | 857.6 | 579.0 | 5111.4 |
| $M-GL-Bt_{randI}$ | 161.8 | 954.6 | 569.1 | 59.6 | 4373.0 | 1067.4 | 1068.8 | 734.5 | 5422.6 |
| $L-GL-Bt_{0.7}I$ | 173.5 | 1072.1 | 540.0 | 72.6 | 1143.9 | 1633.4 | 1274.1 | 578.7 | 5383.4 |
| $L-GL-Bt_{rand}I$ | 136.3 | 1273.7 | 506.5 | 64.0 | 4112.8 | 1163.7 | 1594.3 | 653.1 | 5453.6 |

# D STATISTICAL HYPOTHESIS TESTING OF RANK CORRELATION

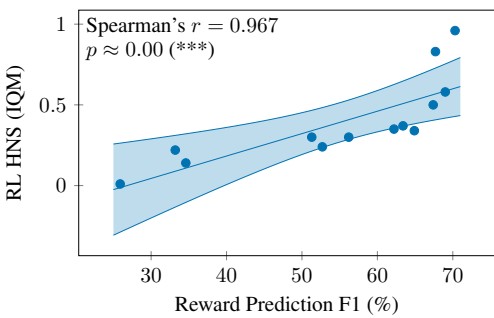 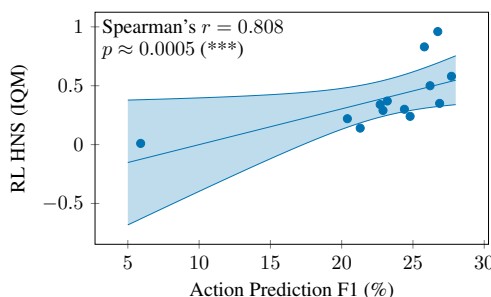

Figure 5: Correlation between the SSL representations' abilities to linearly predict (**Left**) presence of immediate reward and (**Right**) action, versus RL performance using the same representations, measured as the interquartile mean of the human-normalized score (HNS) over 9 Atari games. Each point denotes a separate SSL pretraining method. A linear line of best fit is shown with 95 confidence interval. We compute Spearman's rank correlation coefficient (Spearman's r) and determine its statistical significance using permutation testing (with $n = 50000$). Compared to Fig. 1, we added one extra model which obtained poor probing results to demonstrate that the correlations holds for a wide range of performance levels.

In Fig. 5, we show the correlations results for both the action and reward predictions. We estimate Spearman's rank correlation coefficient (Spearman's r) between the linear probing performance and the (interquartile) mean RL human-normalized score (HNS) over 9 Atari games. The reason for using Spearman's r instead of the Pearson correlation coefficient is because we are interested in whether the relative *ranking* of the models on the linear probing tasks is indicative of the relative ranking of the same models when RL is trained on top of it. As an example, this allows us to say if model A out-ranks model B in the reward prediction task, an RL model trained on top of model A's representations will likely out-perform an RL model trained on top of model B's representation. However, it does not let us predict by how much model A will out-perform model B.

Let $d$ denote the difference in ranking between the linear probing performance and the RL performance, Spearman's r (denoted as $\rho$ below) is computed as,

$$\rho = 1 - \frac{6 \sum_{i=1}^{n} d_i^2}{n(n^2 - 1)}, \tag{1}$$

where $d_i$ is the difference in ranking for the i-th model, and $n$ is the total number of models we have.

We perform statistical hypothesis testing on $\rho$ with null hypothesis $\rho = 0$ (no correlation between linear probing performance and RL performance) and alternative hypothesis $\rho > 0$ (positive correlation). The null distribution is constructed nonparametrically using permutation testing: we sample random orderings of the observed linear probing performance and RL performance independently and compute $\rho$. This is repeated 50,000 times to generate the null distribution (which is centered at $\rho = 0$ as we do not expect randomly ordered values to be correlated). We then compare our observed $\rho$ to this distribution and perform one-tailed test for the proportion of samples larger than our observed $\rho$ to report our p-value.

## D.1 RANK CORRELATION ON A DIFFERENT DATASET

In Fig. 1, we explored the correlation between the RL performance and the reward probing task, where the dataset used for the reward probing was a set of quasi-random trajectories from the DQN dataset, coming from very beginning of the training run of the DQN agent used to collect the data. It is natural to ask whether the correlation results we obtain are sensitive to the specific dataset used. To put this question to the test, we re-run the same reward probing task, this time on the "expert" dataset, i.e. the last trajectories of the DQN dataset, corresponding to a fully trained agent. The results are shown in Fig.6. The Spearman's correlation coefficient that we obtain is the exact same as the one for

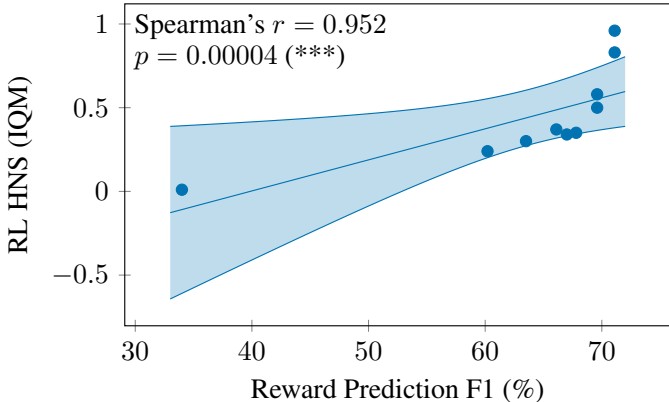

Figure 6: Correlation between SSL representations' abilities to linearly predict presence of immediate reward on a different probing dataset (expert trajectories instead of random ones), for the set of models using Barlow and BYOL objectives.

the random trajectory dataset (even though the reward statistic are different, see Table 14), showing that the correlation result is not sensitive to the probing dataset used.

### D.2 CONFIDENCE INTERVAL OF RL PERFORMANCE AS A FUNCTION OF INDEPENDENT RUNS

We further show the confidence interval of the estimated mean RL performance as the number of independent runs increase. From our total of 10 independent runs each game, we sample with replacement $k \leq 10$ runs ($k$ being number of independent runs we "pretend" to have instead of the full 10), independently for each game. We can compute the IQM over this sample to get an estimate for the IQM as if we only have $k$ independent runs. We repeat this process 10,000 times to construct the 95 confidence interval of the empirical IQM for different $k$'s. Illustrative examples of how much this confidence interval shrinks for different pairs of models is shown in Fig. 7.

We observe in Fig. 7 the mean RL performance estimates have CIs that eventually separate with many independent runs. This is an unbiased but high variance and computationally intensive estimator of the true expected RL performance. On the other hand, the reward prediction F1 score is a computationally cheap, low variance and accurate estimator of the relative model ranks in mean RL performance. This further corroborates our previous results of positive correlation between reward prediction F1 score and mean RL performance (Fig. 1).

### E COMPARISON WITH DOMAIN SPECIFIC PROBING BENCHMARKS

Table 13: Comparison of the correlation of domain specific and domain agnostic (reward prediction) probes with the RL performance.

|  | Spearman's $r$ | $p$ |
| --- | --- | --- |
| AtariARI | 0.527 | 0.058 |
| Reward (ours) | 0.782 | 0.003 |

One of the key advantages of our probing method is that it is domain agnostic, unlike the previously proposed AtariARI benchmark (Anand et al., 2019) which acquires probing labels through the RAM state of the emulator, making their method impractical for image-based trajectories.

To better understand how our probing metrics compare with the domain specific ones in terms of correlations with RL performances, we perform the AtariARI probing benchmarks using our pretrained encoders on the 4 overlapping games (Boxing, Seaquest, Frostbite, DemonAttack) used in both works. For AtariARI, we first calculate the average probe F1 scores across categories, then average this quantity across the games. For reward probing, we apply our own protocol detailed in

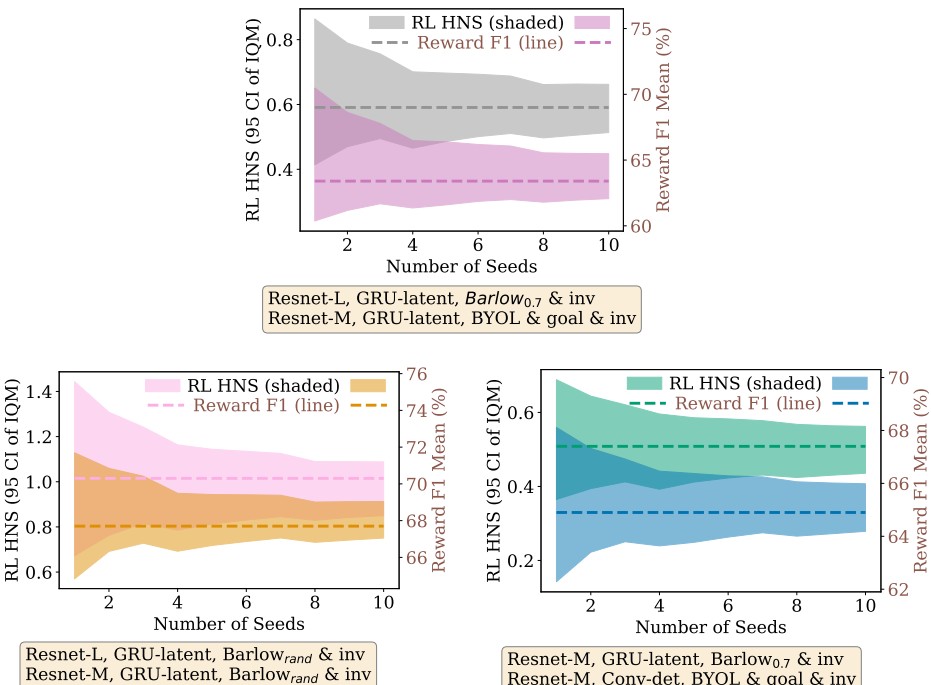

Figure 7: Illustrative examples of 95 confidence interval (CI) of the interquartile mean estimate (IQM) of the human normalized score (HMS) for different models. The CI shrinks as a function of the number of independent runs. CI is estimated using 10,000 bootstrapped samples with replacement. Model names are given below each figure, note the figure colors matches the colors in table 3. The top figure is the same as Fig. 1 (Right). For all cases, the reward prediction F1 score gives us an accurate low-variance estimate for how the two models rank relative to one another.

section 5.1. For RL performance we use the IQM. We report the correlation between the probing metrics and RL performances across different models.

Our results are summarized in Table 13. We find that the correlation between the average probing F1s and RL performances is stronger for our reward probing method. In particular, our probing method has a significant correlation with RL performances ($p < 0.05$), while the AtariARI probing method does not.

## F  PROBING DURING TRAINING

We show evolution of probing performance as training progresses in figure 8.

## G  REWARD STATISTICS IN PROBING DATASETS

In table 14, we report the percentage of states that have a non-zero reward in each of the 9 games, for two different subsets of data:

- Checkpoint 1, which correspond to quasi-random trajectories from the beginning of the training process of DQN. This is the data used for the reward probing in Fig 1.

- Checkpoint 50, which is the last checkpoint of the DQN replay dataset, and corresponds to the fully trained DQN agent, that we assimilate to an expert agent. This data is used for action probing, and for reward probing in Fig.6

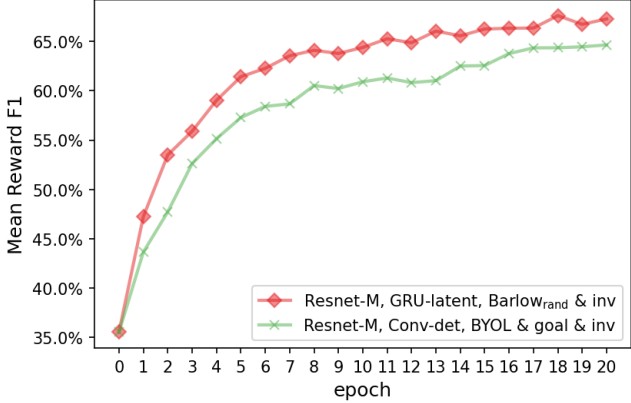

Figure 8: Average reward probing F1s for two SSL setups during different training epochs. Epoch 0 constitutes an untrained model.

Table 14: Percentages of positive rewards in checkpoints 1 and 50 of the DQN replay dataset for 9 games. Checkpoint 1 is used for reward probing and checkpoint 50 is used for expert action probing.

| Game | Ckpt 1 % | Ckpt 50 % |
|---|---|---|
| Amidar | 2.7 | 5.2 |
| Assault | 3.6 | 6.8 |
| Asterix | 5.0 | 6.0 |
| Boxing | 3.5 | 9.3 |
| DemonAttack | 2.1 | 4.7 |
| Frostbite | 4.2 | 2.9 |
| Gopher | 2.8 | 8.5 |
| Krull | 13.2 | 41.7 |
| Seaquest | 1.5 | 7.5 |

All the games have a fairly small percentage of positive reward states, and we generally observe a higher percentage of reward in checkpoint 50, which is expected since the agent is more capable by then.

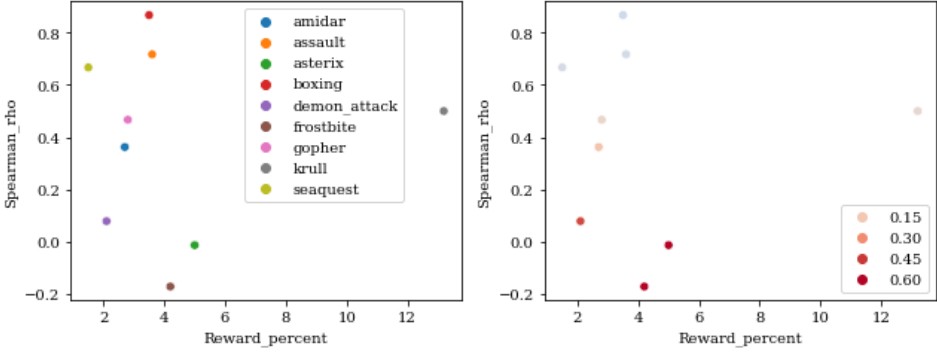

Figure 9: Left: Spearman's correlation coefficient (for the set of models using Barlow and BYOL objectives) between the RL performance on each individual game and the reward probing F1, plotted as a function of the percentage of rewards observed in this game. Right: p-values associated with each of the Spearman's coefficients.

### G.1  IMPACT OF SPARSITY ON THE CORRELATION

In Fig.9, we plot the Spearman's correlation coefficient between the RL performance on each individual game and the reward probing F1, as a function of the percentage of reward observed in each game (see Table 14). We do not observe any particular pattern with respect to the sparsity, suggesting that the probing task is not very sensitive to the sparsity level of each individual game. Note however that, as usual in the Atari benchmark, it is difficult to draw conclusion from any given individual game, and the statistical significance of our results only emerge when considering the set of games as a whole. Indeed, only 3 games achieve individual statistical significance at $p < 0.01$ (Boxing, Seaquest and Assault), while the other do not obtain statistically significant correlations.

## H  LIMITATIONS

One limitation of the current work is that for the presented probing methods to work one needs a subset of the data either with known rewards, where ideally rewards are not too sparse, or with expert actions. If none of the two is available, our method cannot be used. For the reward probing task, the usefulness of the method also depends on the hardness of the reward prediction itself. If the prediction task is too easy, for example because there are rewards at every step, or because the states with rewards are completely different than the ones without (such that even a randomly initialized model would yield features allowing linear separation between the two types of states), then the performance of all the models on this task are going to be extremely similar, with the only differences coming from random noise. In such a case, the performance of the prediction task cannot be used to accurately rank the quality of the features of each of the models.

For future work we also would like to extend the findings of this paper to more settings, for example different environments.

