# OpenReview forum: "Light-weight probing of unsupervised representations for Reinforcement Learning"
_ICLR.cc/2023/Conference — Submitted to ICLR 2023_

### Official Review · Reviewer_vEah · 2022-10-23

**Confidence:** 4
**Correctness:** 4
**Technical Novelty And Significance:** 4
**Empirical Novelty And Significance:** 3
**Recommendation:** 6

**Clarity, Quality, Novelty And Reproducibility:**

Their motivation, model design to evaluate, and evaluation are clearly written except for some minor things, such as the equation typos. Their investigation is novel and looks reproducible through the hyperparameters shared in Appendix.

**Strength And Weaknesses:**

Strength
- The motivation behind this paper is good. Applying the pretrained encoders to RL tasks has been investigated [1,2]. It requires lots of resources. This investigation could be a piece of good evidence to skip the costly evaluation.
- The many variations and ablations are evaluated. For transition modeling, deterministic recurrent modules and RSSM are validated. For SSL, BYOL and Barlow Twins are tested with various configurations. The ablation studies are reported with and without each objective, such as inverse dynamics modeling and goal-conditioned RL.

Weaknesses
- They only investigated the SSL methods, not other unsupervised methods such as VAE. I expected they would cover the overall methods from their title, but it is not.
- They only evaluated nine games. It could not be enough to back up the conclusion.
- In the equation for reward-reg loss in Reward Probing in section 3.2, should the encoder get $o_{t+1}$ not $o_t$? Because the reward $r_t$ is given when the action is given on the observation $o_t$.
- In the equation for action-classif loss in Action Prediction in section 3.2, shouldn't the input be $o_t$ and $o_{t+1}$? In Figure 2, the consecutive observations are given to the loss, but not in this equation.
- In Figure 2, I cannot understand the below sentence. Why is the stacked observation related to data augmentation?
    - > The observations consist of a stack of 4 frames, to which we apply data augmentation before passing them to a convolutional encoder.
-  > The action is represented as a 2D one-hot vector and appended to the input to the first convolutional layer.
    - Why is the action represented through a 2D one-hot vector? The action space is larger than the 2 Dimension.
- For RSSM, you used the discrete latent. Why didn't you try the continuous latent version [3]? Perhaps, because the discrete version outperforms the continuous latent version in [4], but the discrete latent variable training is more unstable than the continuous latent variable training, so maybe the RSSM with continuous latent variable could outperform the discrete latent version.
- In BYOL loss equation in section 4.2, should $q(\hat{y}_{t+k})$ be $q(\hat{e}_{t+k})$?
- For Algorithm 1, why did you use the Pseudo code block? It is just a single equation.
- For the goal-oriented RL loss, please introduce it roughly in the main paper even though the details are in Appendix.
    - In A.7, there are typo $\tilde{e}_t+1$.
- > Similar to the BYOL model, the Barlow model can also be improved with inverse dynamics modeling, while the addition of goal loss has a slight negative impact.
    - It is interesting. Could you analyze this?

[1] Schwarzer, Max, et al. "Pretraining representations for data-efficient reinforcement learning." Advances in Neural Information Processing Systems 34 (2021): 12686-12699.

[2] Dittadi, Andrea, et al. "The role of pretrained representations for the ood generalization of rl agents." arXiv preprint arXiv:2107.05686 (2021).

[3] Hafner, Danijar, et al. "Dream to control: Learning behaviors by latent imagination." arXiv preprint arXiv:1912.01603 (2019).

[4] Hafner, Danijar, et al. "Mastering atari with discrete world models." arXiv preprint arXiv:2010.02193 (2020).

**Summary Of The Paper:**

This paper investigates whether or not light-weight probings to the action between continuous states and reward can measure the pretrained encoder performance on RL tasks. To do this, they pretrained the encoders through self-supervised learning loss with a transition model implemented through recurrent module or recurrent state space modeling with several variations. They tested the pretrained encoder for linear probing and RL tasks. They showed the correlations between the performances on linear probings and RL tasks. They found the linear probing to reward is highly correlated with performance on RL tasks, while the relationship between action linear probing and RL performance is weaker.

**Summary Of The Review:**

This paper investigates the linear probing to the action, and reward could be an indicator of RL performance. Mainly they evaluated the SSL methods, BYOL, and Barlow Twins for 9 Atari tasks. It is interesting and could be helpful for others because applying the pretrained encoder to RL tasks is one of the ways to improve the sample efficiency on RL, but the evaluation is very expensive. However, I felt their writing is rushed, so I think minor revision is necessary.

---

> ### Author Response · Authors · 2022-11-18
> **Response to Reviewer vEah**
>
> We thank the reviewer for the comments. We are happy to find that the reviewer confirms that evaluating standard RL requires lots of resources and our proposal is a step towards skip the costly evaluation.
>
> ## More setups
> > They only investigated the SSL methods, not other unsupervised methods such as VAE. I expected they would cover the overall methods from their title, but it is not.
>
> We ran additional experiments with ATC, ST-DIM and VAE. The results remain statistically significant, correlation coefficient (Spearman’s rho) increased from 0.933 to 0.958. See the global response and the updated paper for additional information
>
> > They only evaluated nine games. It could not be enough to back up the conclusion.
>
> We picked a set of illustrative games. While it would be nice to have more games we tried to strike a balance between comprehensiveness of methods vs. comprehensiveness of games with the amount of compute we have available.
> We note that the results are statistically significant, providing strong backing for the conclusion.
>
> # Questions about the equations
> > Equation for reward-reg loss in Reward Probing in section 3.2. Should the encoder get $o_{t+1}$ not $o_t$? Because the reward $r_t$ is given when the action is given on the observation $o_t$.
>
> $r_t$ is the reward that occurs when executing action $a_t$ in state $s_t$ and reaching $s_{t+1}$. If we give $o_t$ as input (we recall that $o_t$ is the concatenation of the 4 previous frames, it’s a predictive task (“what will happen next”), which is what we want. If we gave $o_{t+1}$ it would not be a hard task since the observation contains the previous states, so it is very easy for a model to figure what happened.
>
> > Equation for action-classif loss in Action Prediction in section 3.2. Shouldn’t the input be $o_{t}$ and $o_{t+1}$ ? In Figure 2, the consecutive observations are given to the loss, but not in this equation.
>
> We only require the “online encoder” to get the representation for observation $o_t$. Thus only $o_t$ is required for action classification. Similarly to the reward case, we want it to be a prediction task (to check whether the expert policy is in the state of functions that the model can learn).
>
> > Figure 2. Why is the stacked observation related to data augmentation?
>
> Observation stacking is not related to data augmentation. It is a standard technique to resolve partial observability. We apply standard observation stacking as is done in the original DQN paper [1], the SGI paper [2], and so on.
>
> [1] Mnih, Volodymyr, et al. "Human-level control through deep reinforcement learning." nature 518.7540 (2015): 529-533.
>
> [2] Schwarzer, Max, et al. "Pretraining representations for data-efficient reinforcement learning." Advances in Neural Information Processing Systems 34 (2021): 12686-12699.
>
>
> > (Section 4.1) “The action is represented as a 2D one-hot vector…” Why is the action represented through a 2D one-hot vector? The action space is larger than the 2 Dimension.
>
> We clarified this in section 4.1:
> The action is represented as a one-hot encoding spatially replicated (in a 2D map) and concatenated with the representation input along the channel dimension.
>
>
> > For RSSM, you used the discrete latent. Why didn't you try the continuous latent version [3]? Perhaps, because the discrete version outperforms the continuous latent version in [4], but the discrete latent variable training is more unstable than the continuous latent variable training, so maybe the RSSM with continuous latent variable could outperform the discrete latent version.
>
> We have no experimental evidence pointing at instability for the discrete latent. As such, we used it since experimental evidence in the papers that the reviewer pointed out indicates that it is better suited for Atari games.
>
> > In BYOL loss equation in section 4.2, should $q(\hat{y}_{t+k})$
>   be
> $q(\hat{e}_{t+k})$?
>
> As explained in the text, $y$ is a down-projection of $e$. We refer to the BYOL paper for explanation on why this projector is required for convergence of the SSL method.
>
> > For Algorithm 1, why did you use the Pseudo code block? It is just a single equation.
>
> We felt this formatting decision was more fitting for the pytorch-style pseudocode (e.g. with operations such as “.detach()” which is not a mathematical operator).
>
> ## Additional clarifications
> > For the goal-oriented RL loss, please introduce it roughly in the main paper even though the details are in Appendix.
>
> Thank you for the suggestion, we have done this in our updated paper.
>
>
> > Can you analyze “Similar to the BYOL model, the Barlow model can also be improved with inverse dynamics modeling, while the addition of goal loss has a slight negative impact.”
>
> We agree this is an interesting observation. While we can have speculative hypotheses for why, to answer this thoroughly requires a degree of work that is beyond the main focus of this paper.

---

### Official Review · Reviewer_etov · 2022-10-24

**Confidence:** 3
**Correctness:** 2
**Technical Novelty And Significance:** 1
**Empirical Novelty And Significance:** 2
**Recommendation:** 3

**Clarity, Quality, Novelty And Reproducibility:**

I think the paper could benefit a lot from improving clarity and presentation. I would especially suggest that authors more explicitly specify the novel contributions and the scope of the work. Maybe the authors could provide more detail in section 5.4. It is possible that I missed parts when reading the manuscript but I believe novelty and originality are limited in light of my comments above.


**Strength And Weaknesses:**

Strengths

1. Identification of an important problem setup in RL, that is how to assess which representations and pretraining data is best suited for improving downstream RL performance.
2. Extensive analysis of various design choices of the model architecture studied

Weaknesses
1. It is not very clear what the goal or objective of the paper is. Authors say they propose an evaluation protocol for unsupervised RL representations that saves up to 600x computation cost. However, this computational cost saving and protocol seems to be not addressed or described in detail in the main paper and doesn't appear to be the main focus of the paper. Instead, the authors analyse a very specific class of self-predictive (recurrent) representation models that are being trained with SSL objectives. I would have expected to see a more thorough coverage of different unsupervised representation learning methods and more empirical analysis to support these claims (see additional comments in point 3)
2. It is not very clear what is actually novel about the proposed model and what is based on prior work. The listed contributions are very vague. I would hope the authors can clarify what exactly their contributions are. Also, there exist additional prior works that very extensively studied various light-weight probing tasks on unsupervised representations and how their performance relates to RL performances [1,2]
3. I am a bit confused by the experiments, especially the correlative analysis and it is not very clear to me if this holds beyond the very particular method and environments. In particular Figure 3 is very confusing and I do not fully understand what the 7 representative setups are supposed to be; how this relates to the 9 rows/models presented in Figure 3 (where do they come from?); and how this relates to the 7 Atari games studied in Figure 1. Without this additional information, it is not clear to me if it is sound to draw a general conclusion about the correlation between probing task performance and RL performance or if there are any other confounders. Also, can authors provide error bars for results in Table 1-5?
4. Minor: There appears to be a slight mismatch between the title/abstract and the presented experiments and contributions of the main paper. The authors study a very particular narrow setting but the title suggests a generally applicable evaluation protocol for different unsupervised representation learning methods.
5. Regarding the two probing tasks I wonder how generally applicable they really are. E.g. the reward task seems to be constrained to have labelled data from a very early point in RL training (i.e. rather random policy), whereas the action prediction is limited to labelled data from close to expert trajectories. To be generally applicable I would have hoped to see both probes being trained on the same labelled dataset to compare apples to apples.

[1] Dittadi, Andrea et al. “The Role of Pretrained Representations for the OOD Generalization of Reinforcement Learning Agents”, ICLR 2022.

[2] Higgins, Irina, et al. "Darla: Improving zero-shot transfer in reinforcement learning." International Conference on Machine Learning. PMLR, 2017.


**Summary Of The Paper:**

This paper attempts to propose an evaluation protocol for lightweight probing of unsupervised representations and investigates the correlation between RL performance and linear probing from a pretrained representation. Authors are testing this on a very specific class of self-predictive (recurrent) representation models that are being trained with SSL objectives. The authors extensively analyse several design choices of their studied model regarding performance on the two probing tasks of predicting reward or action from a held-out labelled train set. Finally, the paper aims to present some correlation between probing and RL performance on 9 Atari games.


**Summary Of The Review:**

While this paper attempts to address an important challenge in RL and aims to propose a generally applicable evaluation protocol for unsupervised pretrained RL representations I am not fully convinced that the paper holds up to these promises and several claims in the paper. I am willing to change my opinion in case I missed central parts of the paper but want to encourage the authors to further improve the manuscript based on some suggestions and comments above and recommend rejection.

---

> ### Author Response · Authors · 2022-11-18
> **Response to Reviewer etov (1/2)**
>
> We thank the reviewer for their time and feedback. We are glad the reviewer found the problem we studied "important" and our analysis "extensive".
>
> ## Contribution of the paper
> > “ not very clear what the goal or objective of the paper is”
>
> > “The listed contributions are very vague”
>
> As stated in the contributions section of our introduction, we focus on providing a widely applicable framework for evaluating representations on the context of RL. Previously, the go-to method was to run full blown RL evaluation (use the pre-trained backbones in a model-free training). This requires running 10 seeds on each of the 27 games, where each seed takes on average 10 hours (as explained in our experimental section). That adds up to 10*27*10 = 2700 gpu hours to complete the evaluation, which is not a practical nor ecological way to iterate over design choices of the SSL method.
>
> Instead, we propose a simple linear probing method, which only requires fitting a linear head to predict some quantity of interest (the instantaneous reward or the action of an expert action). This costs 10 minutes of computation per game, hence a total cost of 27 * 10/60 = 4.5 gpu hours. The total saving is 2700 / 4.5 = 600 as claimed.
>
> The main takeaway is that our evaluation scheme strongly correlates with the “true” performance of the model, proving that this is indeed a viable way to evaluate models.
>
> The contributions we make on the modeling side are incremental in the sense that they combine known architectures/losses from existing papers.
>
> ## Related Work
> > there exist additional prior works that very extensively studied various light-weight probing tasks
>
> We thank the reviewer for the references, we added some discussion about them in our related work.
>
> As for [1] and [2], their probing tasks are either task-specific (predicting some values about the state such as absolute position of objects, types of objects and the surroundings) or method specific (such as the ELBO). By contrast, we strive for a generally applicable probing framework without any assumptions on the method or task. We also refer to our Appendix E, where we tested whether predicting game-specific variables is viable: we found that their correlation is weaker than with our proposed method.
>
> ## Correlative experiments
>
> >it is not very clear to me if this holds beyond the very particular method and environments
>
> We thank the reviewer for the insight. In response to the feedback, we've evaluated our methods on 3 additional pretraining methods from the RL SSL literature: VAE, ST-DIM, and ATC. We find that the correlation coefficient (spearman's rho) is even higher between RL performance and probing F1s when the 3 additional models are introduced, which supports our claim that the probing benchmark is a cost-effective and robust way to evaluate visual representations.
>
> We find that Atari is a challenging benchmark, and due to considerable computational cost we cannot afford to run it on other control tasks. Due to the variety of the games in the benchmark, we expect our findings to apply to other discrete control environments.
>
> ## Question about figure 3 (visualization of the latent)
> > In particular Figure 3 is very confusing
>
> Figure 3 is meant to give some insights on the role of latent variable in the forward model. Beyond raw improvements on the metrics (both the probing tasks and the RL), we qualitatively show that the latent captures the randomness of the environment, providing insight that it works as intended.
>
> ## Question about “representative setups”
>
> > I do not fully understand what the 7 representative setups are supposed to be; how this relates to the 9 rows/models presented in Figure 3
>
> We apologize for the typo in our manuscript, it should have read “9 representative setups”. The 9 rows in Table 3 correspond to the 9 data points (one datapoint=one model) in figure 1, left. These models correspond to the best performing ones from all the axis of variation we considered (the figure is difficult to read if we include extremely low performing one). See Fig 5 in the appendix for a figure with all the models evaluated.
>
> Thanks to the new experiments we conducted, we now have 12 representative setups (both in table 3 and figure 1)
>
> ## Error bars
>
> > Also, can authors provide error bars for results in Table 1-5?
>
> It is too computationally expensive to pretrain the models several times with different seeds, so following best practices in the computer vision SSL community, we report only one pretraining run. As for the logistic regression used to fit the linear classifier, we measured the std to be on the order of 1e-4

---

> ### Author Response · Authors · 2022-11-18
> **Response to reviewer etov (2/2)**
>
> ## Scope of the paper
>
> >There appears to be a slight mismatch between the title/abstract and the presented experiments and contributions of the main paper. The authors study a very particular narrow setting but the title suggests a generally applicable evaluation protocol for different unsupervised representation learning methods
>
> We hope that the additional results on significantly different unsupervised representation learning methods help convince the reviewer that our claim applies to a broad set of methods.
>
> ## Nature of the reward probing tasks
>
> >Regarding the two probing tasks I wonder how generally applicable they really are.
>
> Broadly speaking, realistic RL environments fall in two broad categories:
> - Risk-averse environments, the prime example of which is self-driving (and many other robotics tasks). These typically involve interaction with the real-world in a human-like way, where risks should be minimized at all cost, and negative rewards (eg crashing the car) pretty much never observed). In this case it is likely that a lot of human trajectory are going to be readily available, provided us with a dataset of expert actions.
> - Game-like environments, such as atari, where interactions are cheap. In this case, it is typically very easy to collect data using pseudo random trajectories (and their associated rewards), but very costly to obtain expert trajectories
>
> The two tasks we propose are meant to cover each of the scenarios: In the first case, expert trajectories are readily available, and hence action probing is viable. In the second case, rewards are plentiful and thus reward probing is viable. We are not aware of any non-toy situation which doesn’t fall into either category, and would be very interested if the reviewer has some examples in mind.
>
> > I would have hoped to see both probes being trained on the same labelled dataset to compare apples to apples.
>
> We point the reviewer to appendix D.1, where we performed the reward probing on the expert dataset (ie apples to apples with the action probing). We found the conclusions to be the same.

---

### Official Review · Reviewer_LabE · 2022-10-25

**Confidence:** 3
**Correctness:** 2
**Technical Novelty And Significance:** 2
**Empirical Novelty And Significance:** 2
**Recommendation:** 3

**Clarity, Quality, Novelty And Reproducibility:**

Overall I thought the presentation was straightforward save a few minor confusions:
Is section 4.1 a contribution of this paper? From the introduction and abstract, I assumed that the paper's main contribution was the evaluation protocol, but it was unclear if the architecture in Figure 2 was adapted from past work or newly introduced for this task. I think this would be helpful to clarify because it would be useful to see the evaluation protocol on multiple kinds of models or on models developed in past work.

**Strength And Weaknesses:**

This paper introduces an interesting idea for the important problem of cost-effective evaluations of visual representations. Currently results are limited to Atari and the evaluation is only performed for a small number of models + self-supervised losses. Because the main goal of this paper is to provide an evaluation protocol that can be used in place of downstream RL performance, it would be helpful to see a much broader range of losses and models as well as more difficult control tasks. It's also unclear how well these evaluation protocols predict downstream performance in the presence of task transfer: e.g., one goal in developing visual representation pre-training methods is to get a good generalizable encoder. Because the evaluation leverages reward information and information about the optimal policy, it doesn't seem like it would predict the fitness of an encoder for new tasks.

**Summary Of The Paper:**

This paper develops an evaluation protocol for unsupervised visual pretraining. They learn linear probes to predict expert agent actions and rewards from encoded states. These probes provide a more cost-efficient way of comparing visual representation learning methods for RL. The evaluation protocol is tested on a handful of Atari tasks and the authors show that the performance of networks on the linear probes well correlates with RL performance.

**Summary Of The Review:**

The paper tackles an interesting problem, but I think it should include evaluation over a larger set of

---

> ### Author Response · Authors · 2022-11-18
> **Response to Reviewer LabE**
>
> We thank the reviewers for the feedback and suggestion, and we're glad that the reviewer thinks the paper "introduces an interesting idea for the important problem of cost-effective evaluations of visual representations."
>
> ## Broader range of losses and models
> > Currently results are limited to Atari and the evaluation is only performed for a small number of models + self-supervised losses. Because the main goal of this paper is to provide an evaluation protocol that can be used in place of downstream RL performance, it would be helpful to see a much broader range of losses and models as well as more difficult control tasks.
>
> In response to the feedback, we've evaluated our methods on 3 additional pretraining methods from the RL SSL literature: VAE, ST-DIM, and ATC. We find that the correlation coefficient (spearman's rho) is even higher between RL performance and probing F1s when the 3 additional models are introduced, which supports our claim that the probing benchmark is a cost-effective and robust way to evaluate visual representations.
>
> We find that Atari is a challenging benchmark, and due to considerable computational cost we cannot afford to run it on other control tasks. Due to the variety of the games in the benchmark, we expect our findings to apply to other discrete control environments.
>
> ## Whether probing metric from one task generalizes to other tasks
> > It's also unclear how well these evaluation protocols predict downstream performance in the presence of task transfer: e.g., one goal in developing visual representation pre-training methods is to get a good generalizable encoder. Because the evaluation leverages reward information and information about the optimal policy, it doesn't seem like it would predict the fitness of an encoder for new tasks.
>
> While it is an interesting question how such task specific probing metrics reflect the representations fitness for other tasks, evaluating the “universality” of the representation is out of the scope of our work. We do note that our framework would nonetheless help in this case: given a new task -- i.e. a new reward function -- one could do linear probing of reward prediction (learning state-reward pairs with the new reward function) more efficiently than training an RL agent to maximize the same reward function (which requires learning state-value functions). Similarly, if expert action exists for the new task we can efficiently action-probe the representations. In both cases, our method provides a net benefit compared to the only alternative to date, which is to perform full-blown RL evaluation, which is the crux of our argument.
>
> ## Contributions
> > Is section 4.1 a contribution of this paper? From the introduction and abstract, I assumed that the paper's main contribution was the evaluation protocol, but it was unclear if the architecture in Figure 2 was adapted from past work or newly introduced for this task. I think this would be helpful to clarify because it would be useful to see the evaluation protocol on multiple kinds of models or on models developed in past work.
>
> The architecture in Figure 2 is the same as SGI's pretraining framework. However, we introduced variations to this architecture, including different choices for Distance function and dynamic models, a joint embedding setting where the online and target branches are identical, and a setting where the target branch is a frozen random network. Using our proposed probing benchmark as a guide, we were able to rapidly evaluate these different setups, and make significant improvements over the SGI baseline.
>
> We also added results for other unsupervised methods based on contrastive and reconstruction losses, and found strong correlation between RL and probing performances for those ones as well. See the main response and the updated paper.

---

### Official Review · Reviewer_Vgoa · 2022-10-25

**Confidence:** 3
**Correctness:** 3
**Technical Novelty And Significance:** 3
**Empirical Novelty And Significance:** 3
**Recommendation:** 6

**Clarity, Quality, Novelty And Reproducibility:**

The paper is clearly written, the experiments are carefully done and interesting ablations are conducted.  Although linear probing is common in computer vision representation evaluation, the generalization to RL and reward prediction is novel as far as I am aware.

**Strength And Weaknesses:**

The paper tackles a very difficult and relevant problem, that of evaluating self-supervised representations.  The paper shows evidence that linear probing can give strong indications of eventual RL training performance, which promises to shorten evaluation time and could be impactful in the representation learning for reinforcement learning field.  My main concern with the paper is the lack of diversity in methods used to assess the correlation between linear probes and RL training performance.  All methods compared are ablations of the self-predictive representation approach described in the paper.  While these are important and elucidating experiments, I would like to see a broader set of methods compared, like augmentation-based representations (DrQ or CURL).  Do these correlations hold in these cases as well?  Also, I'm curious about the noise in the linear probe F1 score.  Do the numbers reported in the tables stay the same regardless of random seed?

**Summary Of The Paper:**

This paper proposes a method for evaluating unsupervised representation learning in reinforcement learning.  Using a linear probe on top of frozen, pretrained representations, the paper suggests learning to predict reward values from various states in downstream tasks.  Additionally, the paper uses a linear probe to predict expert actions from learned representations.  They authors show evidence that, for a selection of representation learning approaches, the F1 score of the linear probe correlates strongly with full reinforcement learning on the downstream task.

**Summary Of The Review:**

The paper presents an interesting approach to an interesting problem, with the promise of helping evaluate representations much more quickly.  Although the ablation studies are thorough, there could be a broader comparison to other styles of representation learning, which would significantly strengthen the claims that linear probing for reward prediction correlates well with downstream RL training.

---

> ### Author Response · Authors · 2022-11-18
> **Response to Reviewer Vgoa**
>
> We thank the reviewer for their time and insightful review. We appreciate that the reviewer found our paper "clearly written", our ablations "carefully done" and our approach "interesting" and "novel".
>
> ## Broader set of experiments
> >  I would like to see a broader set of methods compared, like augmentation-based representations (DrQ or CURL).
>
> We thank the reviewer for the constructive suggestion. DrQ and CURL don’t fit in the setting we are considering, since we are looking at the pre-training -> fine-tuning  paradigm, whereas these papers are looking at additional losses or data augmentations directly during RL training (no pre-training).
>
> Instead, we added results one reconstruction-based method (VAE-t) and two contrastive ones (STDIM and ATC). We find that the rank correlation also holds very strongly with those methods. We hope that this provides additional experimental evidence on top of the joint-embedding (non-contrastive) methods we studied originally. See the overall response and the updated paper for more information.
>
> ## Confidence intervals for F1
> >  Also, I'm curious about the noise in the linear probe F1 score. Do the numbers reported in the tables stay the same regardless of random seed?
>
> It is too computationally expensive to pretrain the models several times with different seeds, so following best practices in the computer vision SSL community, we report only one pretraining run. As for the logistic regression used to fit the linear classifier, we measured the std to be on the order of 1e-4, and added that in the paper.

---

### Author Response · Authors · 2022-11-18
**General comment and additional results**

We thank all reviewers for carefully reading our work and providing useful feedback.
To address the shared concern of reviewers about the applicability of our proposed light-weight evaluation to other existing unsupervised pre-training methods, we ran additional experiments with two contrastive methods: ATC [1], ST-DIM [2]; and one reconstruction-based: VAE [3]. We used the implementations, architectures, and hyperparameters from rlpyt [4] repository. Notably, these methods use a smaller backbone with just three convolutional layers. Thus, the new experiments introduce a variety of both methods and model sizes. With these additional results, we find that the correlation coefficient between reward probing and final performance on Atari 100k is even higher than when only considering experiments with joint-embedding methods (Spearman’s rho is 0.958 with p-value 2-e5, compared to 0.933 with p-value 2e-4).

We add the new results to the paper (figure 1 and table 3) and additionally duplicate them below:
Backbone | Transition | Objectives | Reward | Action | RL Median | RL IQM
--|--|--|--|--|--|----
ResNet-L | GRU-lat | Barlow-rand, inv  | 70.3 | 26.7 | 0.62 | 0.96
ResNet-L | GRU-lat | Barlow-0.7, inv | 69.0 | 27.7 | 0.47 | 0.58
ResNet-M | GRU-lat | Barlow-rand, inv  | 67.7 | 25.8 | 0.38 | 0.83
ResNet-M | GRU-lat | Barlow-0.7, inv  | 67.4 | 26.2 | 0.45 | 0.50
ResNet-M | GRU-lat | BYOL, goal, inv  | 63.4 | 23.2 | 0.32 | 0.37
ResNet-M | GRU-det | BYOL, goal, inv  | 62.2 | 26.9 | 0.31 | 0.35
ResNet-M | Conv-det | BYOL, goal, inv | 64.9 | 22.7| 0.23 | 0.34
ResNet-M | GRU-lat | Barlow-0.7  | 56.2 | 24.4 | 0.09 | 0.30
ResNet-M | Conv-det | Barlow-0.7, goal, inv | 52.7 | 24.8 | 0.12 | 0.24
S | None | ATC **(new)** | 51.3 | 22.9 | 0.22 | 0.30
S | None | ST-DIM **(new)** | 34.6 | 21.3 | 0.05 | 0.14
S | None | VAE-T **(new)** |33.2 | 20.4 | 0.11 | 0.22

### Changes to the paper
Following feedback from the reviewers, here is the summary of changes to the content of the paper:
- Added results with more unsupervised pre-training methods (all reviewers);
- Added missing related work (reviewers etov and vEah);
- Added confidence intervals (reviewers etov and Vgoa);
- Minor clarifications and details (all reviewers).

### References

[1]  Stooke A et al., "Decoupling Representation Learning from Reinforcement Learning", ICML 2021

[2]  Anand A et al., "Unsupervised State Representation Learning in Atari", NeurIPS 2019

[3]  Kingma D P and Welling M, "Auto-Encoding Variational Bayes", 2014

[4]  Stooke A and Abbeel P, "rlpyt: A Research Code Base for Deep Reinforcement Learning in PyTorch", 2019

---

### Decision · Program_Chairs · 2023-01-20

**Decision:**

Reject

**Justification For Why Not Higher Score:**

N/A

**Justification For Why Not Lower Score:**

N/A

**Metareview: Summary, Strengths And Weaknesses:**

The final score of this paper is 6533 (Reviewer vEah mentioned he/she will lower the score from 6 to 5 during discussion).

This paper introduce new evaluation protocol for unsupervised visual representation for RL. It can provide a better method for exploring the pertaining algorithms without running RL extensively. The reviewers generally agree that the paper proposed an interesting idea and the experimental analysis is extensive. However, during the discussion, the reviewers still show concerns on how generally the probing tasks are applicable (beyond the Atari games) and the soundness of the correlative analysis. With this concern most reviewers recommended to reject the paper and AC agrees with the majority of the reviewers.